

# Assessment and phenotypic identification of millet germplasm (*Setaria italica* L.) in Liaoning, China

Xintong Li[1], Weifeng He[2], Honghao Wang[2] and Min Xu[2]

[1] School of Accounting, Guizhou University of Finance and Economics, Guiyang, Guizhou, China
[2] Cash Crop Institute, Liaoning Academy of Agricultural Science, Liaoyang, Liaoning, China

## ABSTRACT

**Aims:** This study evaluated millet germplasms in Liaoning Province to support the collection, preservation and innovation of millet germplasm resources.

**Methods:** The study was conducted from 2018 to 2020, involved the selection of 105 millet germplasm resources from the Germplasm Bank of the Liaoning Academy of Agricultural Sciences (LAAS), the observation and recording of 31 traits, and the application of multivariate analysis methods to assess phenotypic diversity.

**Results:** From the diversity analysis and correlation analysis, it was found that the tested traits had abundant diversity and complex correlations among them. Principal component analysis (PCA) comprehensively analyzed all quantitative traits and extracted seven principal components. Grey relational analysis (GRA) highlighted the varied contributions of different traits to yield. Through systematic cluster analysis (SCA), the resources were categorized into six groups at Euclidean distance of 17.09. K-mean cluster analysis determined the distribution interval and central value of each trait, then identified resources with desirable traits.

**Conclusion:** The results revealed resources that possess characteristics such as upthrow seedling leaves, more tillers and branches, larger and well-formed ears, and lodging resistance prefer to higher grain yield. It was also discovered that the subear internode length (SIL) could be an indicator for maturity selection. Four specific resources, namely, Dungu No. 1, Xiao-li-xiang, Basen Shengu, and Yuhuanggu No. 1, were identified for further breeding and practical applications.

# INTRODUCTION

Millet (*Setaria italica* L.) is a diploid ($2n = 2X = 18$) species of foxtail grass (*Gramineae*, *Setaria*), which originated in China, is one of the world's oldest cultivated crops, with a cultivation history of more than 8,000 years (*Lu et al., 2009*). Millet possesses attributes such as drought resistance, water efficiency, high light utilization capacity, high storage convenience, and dual utility as both a grain and a grass (*Yang et al., 2012*). The millet seed kernel is a reservoir of well-balanced nutrients (*Diao, 2011*), comprising ample proteins (*Liu et al., 2009*) and vitamins (*Liu & Lu, 2013*), is commonly chosen as the primary dietary option for new mothers or recovering patients.

Corresponding author
Min Xu, shumin690101@163.com

According to statistical data, there are more than 40,000 millet germplasm worldwide. *Li et al. (1996)* China possesses the most abundant germplasm resources, the National Crop Germplasm Resource Bank of China houses over 26,000 germplasm resources, accounting for 70% of the global collection (*Liu et al., 2009*; *Doust et al., 2009*). Previous researchers have made significant progress in researching and utilizing millet germplasm resources, these studies primarily focused on genetic diversity analysis of phenotypic traits. For instance, *Wang et al. (2016)* comprehensively evaluated 15 phenotypic traits in 878 millet resources globally and identified eight key indicators, such as leaf sheath color, ear length, seed color, and kernel color, for phenotype identification. *Tian (2010)* investigated the genetic diversity of 482 millet resources in Henan and Shandong provinces and discovered that the diversity level of millet cultivated varieties was considerably lower than that of local varieties, suggesting certain traits of greater significance in the breeding process. *Li et al. (1996)* examined 23,381 Chinese landrace millet samples and conducted a comprehensive analysis of 11 agronomy traits, among these traits, only seedling leaf color, starch composition, and 1,000-grain weight showed significant regional differences in phenotypic diversity indices. In addition, molecular biotechnology has been applied in the study of millet germplasm resources. For instance, *Yang et al. (2003)* and *Schontz & Rether (1998)* identified significant genetic diversity in millet resources from various geographic regions. *Jia et al. (2013)* conducted genome resequencing to analyse the diversity of 916 core millet germplasms, providing insight into the geographical distribution of millet genetic resources.

As stated by *Liu et al. (2019)*, the primary millet-producing regions in China are concentrated in the northern and eastern areas. Liaoning, as a significant millet-producing region in China, is renowned for its golden red grain color, pleasant taste, and high-quality grains. Given the rapid exchange frequency and innovation speed of germplasm, there is an urgent need to evaluate and sort out the existing millet resources in Liaoning Province. In comparison to molecular markers, simpler and more intuitive indicators are required to evaluate resource materials in practical production. Therefore, this study was conducted to comprehensively assess the current status of millet germplasm resources in Liaoning Province, screen agronomic traits for a rational evaluation of resource materials, and further identify specific germplasm resources to advance scientific research and production of millet in Northeast China.

## MATERIALS AND METHODS

Portions of this text were previously published as part of a preprint, (https://www.biorxiv.org/content/10.1101/2024.04.14.589429v1).

### Experimental materials

A total of 105 millet germplasm resources preserved in the Germplasm Bank of the Liaoning Academy of Agricultural Sciences (LAAS) were selected for this study. These resources originated from various regions, including Liaoning (52), Beijing (20), Hebei (eight), Jilin (nine), Inner Mongolia (nine), Heilongjiang (two), and Shanxi (five).

Approximately half of the materials are from Liaoning, with the remaining materials sourced from area that north of the Yellow River.

### Experimental design

The experiment was conducted over three growing seasons, from 2018 to 2020, at the experimental site of the Cash Crops Institute of Liaoning (Liaoyang City, Liaoning Province, China). The soil at the site is sandy loam, with the following composition in the topsoil: $1.97 \times 10^3$ mg/kg organic matter, 800 mg/kg total nitrogen, 73.4 mg/kg alkali-hydrolyzable nitrogen, 23.6 mg/kg available phosphorus, and 247.5 mg/kg available potassium. The materials were arranged sequentially during sowing, with each material planted in four rows with 3–4 cm plant spacing. The length of each row was 5 m, with a spacing of 50 cm, resulting in a plot area of 10 square meters. Sowing took place around May 11th, and harvesting was performed around September 25th. A three-compound fertilizer (N:P:K = 15%:15%:15%) was applied as a base fertilizer. Ploughing and weeding were conducted three times during the growth period.

Two points were selected from each plot, and five consecutive plants with uniform growth situation were chosen from each point as samples. Qualitative and quantitative traits were investigated following the Descriptors and Data Standard for Millet (*S. italica* L.) compiled by *Lu (2006)*. Among the traits, 12 quantitative traits and 19 qualitative traits that shown difference across resources. were selected, ignoring traits without difference.

### Data processing

The average and interannual values of the 10 plants were calculated with EXCEL 2007, then analyzed further. Phenotypic diversity analysis and systematic cluster analysis (SCA) were conducted based on all 31 trait, correlation analysis, principal component analysis (PCA), gray relational analysis (GRA), and K-means cluster analysis were performed based on the 12 quantitative traits (*Tang, 2010*). Shannon diversity index of 19 qualitative traits was calculated using the method described by *Wang et al. (2021)*.

## RESULTS

### Phenotypic diversity analysis of millet germplasm resources

Table 1 shows the Shannon diversity index (DI) and distribution frequency of phenotypic characters of 19 quality traits. By comparing the frequency, it was found that tiller habit and branch habit showed weak, moderate, and strong characteristics with a uniform distribution, resulting in a greater diversity index (DI). The phenotypic distribution of the other 17 traits showed clear tendencies, with focused on 1 or 2 characteristics, resulting in a lower DI. It is worth noting that traits related to pigmentation show obvious distribution tendencies, with traits related to leaves mainly being green and traits related to flowers primarily being yellow.

The phenotypic diversity analysis of 12 quantitative traits is presented in Table 2. The coefficient of variation (CV), which represents the degree of trait dispersion, ranged from 6.34% to 43.77% for the 12 traits. The highest CV was observed for stem number per plant,

**Table 1 Phenotypic diversity of 19 qualitative traits.**

| Trait | | Shannon diversity index DI | Distribution frequency | | | | | | |
|---|---|---|---|---|---|---|---|---|---|
| | | | 1 | 2 | 3 | 4 | 5 | 7 | 9 |
| Leaf sheath color | LSC | 0.5257 | 0.7810 | – | 0.2190 | – | – | – | – |
| Leaf color of seedling | LCS | 0.2925 | 0.9143 | 0.0857 | – | – | – | – | – |
| Bristle color | BC | 0.5324 | 0.0762 | 0.8476 | 0.0762 | – | – | – | – |
| Bristle length | BL | 0.7438 | 0.0476 | 0.7524 | 0.1810 | 0.0190 | – | – | – |
| Protecting glume color | PGC | 0.3673 | 0.0095 | 0.8952 | 0.0952 | – | – | – | – |
| Stigma color | STC | 0.6483 | 0.2190 | 0.7524 | 0.0286 | – | – | – | – |
| Anther color | AC | 0.6830 | 0.1524 | 0.0762 | 0.7714 | – | – | – | – |
| Seed color | SC | 0.1075 | – | 0.9810 | – | 0.0095 | 0.0095 | | |
| Kernel color | KC | 0.0943 | – | 0.9810 | – | 0.0190 | – | – | – |
| Seedling leaf attitude | SLA | 0.7825 | 0.0381 | 0.6286 | 0.3333 | – | – | – | – |
| Blooming leaf attitude | BLA | 0.9655 | 0.6667 | 0.1143 | 0.0476 | 0.1714 | – | – | – |
| Tiller habit | TH | 1.0566 | 0.3524 | 0.4381 | 0.2095 | – | – | – | – |
| Branch habit | BH | 1.0543 | 0.3429 | 0.4476 | 0.2095 | – | – | – | – |
| Peduncle shape | PS | 0.9367 | – | 0.2190 | 0.6095 | 0.1714 | – | – | – |
| Ear compactness | EC | 0.5004 | 0.0286 | 0.8476 | 0.1238 | 0.0000 | – | – | – |
| Spike density | SD | 0.5799 | 0.7333 | – | 0.2667 | 0.0000 | – | – | – |
| Ear shape | ES | 1.1206 | 0.0381 | 0.4476 | 0.3810 | 0.1333 | – | – | – |
| Shattering habit | SH | 0.7091 | – | 0.3714 | 0.6190 | 0.0095 | – | – | – |
| Lodging resistance | LR | 1.0316 | – | – | 0.4762 | – | 0.4000 | 0.1048 | 0.0190 |
| Mean | | 0.6701 | 0.3143 | 0.5554 | 0.2905 | 0.0593 | 0.2048 | 0.1048 | 0.0190 |

Note:
Arabic numbers (1–9) in the first line of the table refer to the phenotypic trait according to Descriptors and Data Standard for Millet (*S. italica* (L.)) (*Lu, 2006*).

**Table 2 Phenotypic diversity of 12 quantitative traits.**

| Traits | | | Min V. | Max V. | Mean V. | SD | CV % | Shannon diversity index DI |
|---|---|---|---|---|---|---|---|---|
| Growing period d | | GP | 101.0 | 131.0 | 118.8095 | 7.5359 | 6.34 | 1.6966 |
| Stem | Stem number per plant | SNP | 1.0 | 3.0 | 1.7714 | 0.7753 | 43.77 | 1.0566 |
| | Main stem length cm | MSL | 90.0 | 170.0 | 133.8476 | 19.0252 | 14.21 | 2.0428 |
| | Main stem diameter mm | MSD | 15.0 | 41.0 | 25.8952 | 4.2357 | 16.36 | 1.8507 |
| | Main stem nod number | MSN | 0.3 | 1.0 | 0.5224 | 0.1506 | 28.83 | 1.5176 |
| Ear | Peduncle length cm | PL | 9.0 | 16.0 | 13.1154 | 1.4300 | 10.90 | 1.7074 |
| | Main ear length mm | MEL | 16.0 | 40.0 | 22.8476 | 4.0187 | 17.59 | 1.8741 |
| | Main ear diameter mm | MED | 0.7 | 4.2 | 2.6352 | 0.5036 | 19.11 | 1.7649 |
| Yield | Grass weight per plant g | GWP | 9.2 | 119.8 | 47.0400 | 21.5636 | 45.84 | 1.9513 |
| | 1,000-seed weight g | TSW | 2.0 | 3.9 | 2.6886 | 0.3501 | 13.02 | 1.5784 |
| | Ear weight per plant g | EWP | 11.5 | 41.1 | 22.5114 | 5.9484 | 26.42 | 1.9967 |
| | Seed weight per plant g | SWP | 9.3 | 27.3 | 15.5143 | 4.1043 | 26.45 | 1.9570 |

**Table 3 Correlation analysis of 12 quantitative traits.**

| Correlation coefficient | SNP | MSL | PL | MSD | MSN | MEL | MED | GP | GWP | TSW | EWP |
|---|---|---|---|---|---|---|---|---|---|---|---|
| SNP | 1 | | | | | | | | | | |
| MSL | 0.0237 | 1 | | | | | | | | | |
| PL | 0.0834 | 0.4284** | 1 | | | | | | | | |
| MSD | 0.1965 | −0.1506 | 0.0828 | 1 | | | | | | | |
| MSN | −0.1693 | 0.0881 | 0.0177 | −0.0759 | 1 | | | | | | |
| MEL | 0.2233* | 0.3255** | 0.1945* | 0.2463* | −0.2151 | 1 | | | | | |
| MED | 0.2622** | −0.0081 | −0.0073 | 0.1809 | −0.0935 | 0.0051 | 1 | | | | |
| GP | 0.4105** | 0.3028** | 0.1997* | 0.0978 | −0.0956 | 0.4000** | 0.1920* | 1 | | | |
| GWP | −0.0107 | 0.0811 | 0.0759 | −0.1945* | 0.2872** | 0.0427 | −0.1058 | 0.0558 | 1 | | |
| TSW | −0.1904 | −0.0446 | −0.0274 | −0.0170 | 0.0525 | −0.0286 | −0.0451 | −0.0242 | −0.0508 | 1 | |
| EWP | 0.3955** | 0.2927** | 0.1821 | 0.0931 | −0.2846** | 0.4612** | 0.2799** | 0.3303** | −0.1609 | −0.0955 | 1 |
| SWP | 0.2533** | 0.2147 * | 0.1777 | 0.0035 | −0.2672** | 0.3762** | 0.2573** | 0.1938* | −0.0921 | −0.0839 | 0.8266** |

**Note:**
* indicates a significant association ($r_{0.05}$ = 0.1918), ** indicates a highly significant correlation ($r_{0.01}$ = 0.2504).

followed by main stem node number, seed weight per plant, and ear weight per plant, indicating a high degree of variation and potential for genetic improvement. On the other hand, the subear internode length and growing period had CVs close to or less than 10%, suggesting lower dispersion and relatively stable performance among the varieties. The Shannon DI, which reflects the distribution of trait performance, ranged from 1.0566 to 2.0428. The main stem length, ear weight per plant, seed weight per plant, and grass weight per plant had DI values close to or greater than 2.0, indicating that these traits performance are scattered and susceptible to external conditions. Conversely, Stem number per plant had the lowest DI value, close to 1.0, indicating that these traits performance are stable, reflecting a simple genetic basis.

## Correlation analysis of millet germplasm resources

Correlation analysis was performed based on the 12 quantitative traits (Table 3). The ear weight per plant showed a significant positive correlation with seed weight per plant, and both two traits were positve correlatad with the main ear length significantly; Main ear diameter and stem number per plant also showed a highly significant negative correlation with main stem nod number. Ear weight per plant had a significant positive correlation with main ear length and growing period, and seed weight per plant showed significant positive correlations with these three traits. These results indicate that grain yield is positively correlated with ear size (main stem length and main stem diameter) and ear setting potential (growing period and stem number per plant). Additionally, grass weight per plant was positively correlated with main stem nod number and main stem diamete, these two traits represent vegetative growth. Growing period, which represents the growth potential of the plant, showed positive correlations with stem number per plant, main stem length, main ear length, peduncle length, and main ear diameter.

**Table 4 Principal components analysis and eigenvalue of 12 quantitative traits of 105 millet resources.**

| Traits | | 1st Principle component | 2nd Principle component | 3rd Principle component | 4th Principle component | 5th Principle component | 6th Principle component | 7th Principle componen |
|---|---|---|---|---|---|---|---|---|
| Stem | GP | −0.5030 | 0.3811 | −0.0159 | 0.6491 | −0.2988 | −0.0706 | 0.2702 |
| | SNP | −0.0550 | 0.1803 | 0.8425 | −0.0217 | 0.2663 | 0.1317 | 0.0463 |
| | MSL | 0.5468 | 0.3448 | 0.1252 | 0.1635 | −0.1777 | −0.5729 | 0.2189 |
| | MSD | 0.3181 | 0.6280 | −0.1494 | −0.0008 | 0.4893 | −0.1614 | −0.1068 |
| | MSN | −0.3772 | 0.4097 | −0.2383 | −0.1026 | 0.4557 | −0.0355 | −0.3653 |
| Ear | SIL | −0.2683 | 0.3836 | −0.0633 | 0.2617 | −0.4812 | 0.5356 | −0.3191 |
| | MEL | −0.6281 | −0.2355 | −0.1166 | 0.1917 | 0.4531 | 0.1703 | 0.4657 |
| | MED | −0.3383 | −0.3324 | 0.0922 | −0.6709 | −0.3168 | −0.2211 | −0.1077 |
| Yield | GWP | 0.5415 | −0.3404 | 0.0828 | 0.5759 | −0.2122 | 0.1108 | −0.1540 |
| | TSW | 0.2867 | 0.2468 | −0.6220 | −0.3887 | −0.2632 | 0.1967 | 0.3184 |
| | EWP | 0.1740 | 0.4581 | 0.4012 | −0.4906 | −0.1878 | 0.4226 | 0.2398 |
| | SWP | 0.6166 | −0.2527 | −0.1100 | 0.0748 | 0.4040 | 0.5010 | 0.0581 |
| Eigen value | | 3.1149 | 1.6634 | 1.2091 | 1.1145 | 0.9708 | 0.9199 | 0.8030 |
| Contribution rate | | 25.9573 | 13.8613 | 10.0760 | 9.2874 | 8.0900 | 7.6662 | 6.6915 |
| Cumulative C.R. | | 25.9573 | 39.8186 | 49.8946 | 59.1820 | 67.2720 | 74.9382 | 81.6297 |
| Factor weight | | 31.7988 | 16.9808 | 12.3435 | 11.3775 | 9.9106 | 9.3914 | 8.1974 |

## Principal component analysis of quantitative traits

The PCA results for 12 quantitative traits from 105 millet germplasms are presented in Table 4. Seven principal components with eigenvalues greater than or close to one were selected, which collectively contributed to 81.63% of the variance and encapsulated most of the genetic information of the millet germplasms.

The eigenvector value of each trait indicates its contribution to the principal component (PC). By comparing the eigenvector values, the following observations were made: the 1st PC was primarily associated with main ear length (−), main stem length (+), seed weight per plant (+), grass weight per plant (−), and growing period (−); the 2nd PC was mainly influenced by main ear diamete (+), ear weight per plant (+), and main stem nod number (+); the 3rd PC was primarily influenced by stem number per plant (+), 1,000-seed weight (−), and Ear weight per plant (+); the 4th PC was mainly influenced by MED (−), GP (+), GWP (+) and EWP (−); the 5th PC was mainly influenced by main stem diameter (+), peduncle length (−), main stem nod number (+), main ear length (+) and seed weight per plant (+); the 6th PC was primarily influenced by main stem length (−), peduncle length (+), seed weight per plant (+), and ear weight per plant (+); and the 7th PC was mainly influenced by main ear length (+) and main stem nod number (−). Compared to correlation analysis, PCA provided an interpretation of the correlation between quantitative traits and specifically emphasized the importance of the GP and TSW.

**Table 5 Grey relational analysis of seed yield and 11 quantitative traits.**

| Trait | Interacting coefficient | Correlation sequence | Trait | Interacting coefficient | Correlation sequence |
|-------|-------------------------|----------------------|-------|-------------------------|----------------------|
| EWP | 0.5987 | 1 | PL | 0.4168 | 7 |
| SNP | 0.4674 | 2 | MSD | 0.3915 | 8 |
| MEL | 0.4575 | 3 | GWP | 0.3808 | 9 |
| MED | 0.4332 | 4 | TSW | 0.3714 | 10 |
| GP | 0.4169 | 5 | MSN | 0.3476 | 11 |
| MSL | 0.4168 | 6 | | | |

## Gray relational analysis of quantitative traits and yield

After standardized processing, the seed weight per plant was taken as the reference column, and the correlation between the main quantitative traits and the seed weight per plant was analysed (Table 5). Among the tested quantitative traits, ear weight per plant had the greatest impact on the seed weight per plant, followed by stem number per plant, main stem length, and main ear diameter, these four traits contribute most to the ear-bearing capacity of the plant. The coefficients of growing period, main stem length, and peduncle length were very close, these three traits are mainly related to maturity and main stem growth. The main stem diameter, grass weight per plant, and main stem nod number represent the vegetative growth status of the plant, and their contributions to the seed weight per plant decrease in turn.

## System cluster analysis of all traits

Based on the performance of the 31 studied traits (12 quantitative traits + 19 quality traits) of 105 millet resources, system cluster analysis (SCA) analysis was conducted *via* standardized data transformation-Euclidean distance-deviation square sum method. All materials were divided into six groups at Euclidean distance = 17.09 (Fig. 1).

The 19 quantitative traits for each group were analyzed and compared (Table 6). The 1st group consisted of 11 samples, which exhibited mid-earlier maturity, fewer tillers, taller and slender main stem with multiple nodes, smaller main ears, small seeds with medium grain yield, and higher grass yield. The 2nd group included 30 materials, showing earlier maturation, fewer tillers, lower and slenderer main stem with multiple nodes, smaller main ears, larger seeds but lower grain yield, and higher grass yield. The 3rd group consisted of 33 materials displaying mid-later maturation, medium tiller ability, taller main stem with multiple nodes, medium-sized main ears, larger seeds with higher grain yield, and medium grass yield. The 4th group comprised six materials, showing later maturation, multiple tillering, taller and stronger main stems with multiple nodes, larger main ears, smaller seeds but higher grain yield, and greater grass yield. The 5th group contained five materials, exhibiting mid-later maturation, multiple tillering, lower and stronger main stem with fewer nodes, shorter and thicker main ears, medium-sized grains, lower grain yield and lower grass yield. The 6th group consisted of 20 materials that exhibited mid-later

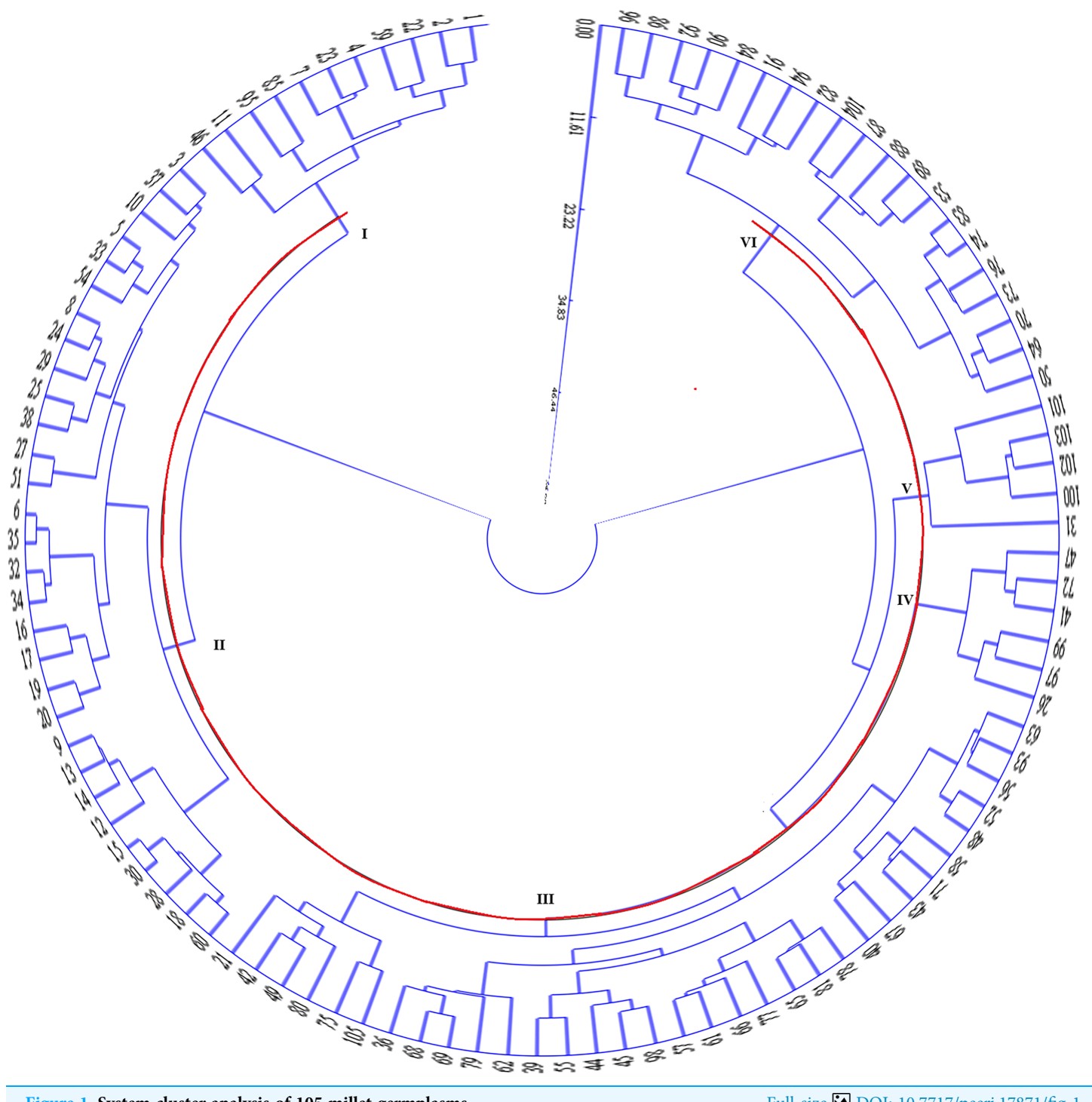

**Figure 1 System cluster analysis of 105 millet germplasms.**

maturation, multiple tillering, taller and slenderer main stems, smaller main ears, smaller seeds but higher grain yields, and lower grass yields.

The 12 quantitative traits were analyzed and compared too (Table 6). The qualities of each group were also compared. The 1st group had unique characteristics in terms of leaf

**Table 6 System cluster analysis of 12 quantitative traits.**

| Traits | | 1st Group | 2nd Group | 3rd Group | 4th Group | 5th Group | 6th Group |
|---|---|---|---|---|---|---|---|
| Stem | GP | 117.0000 | 113.4333 | 121.4848 | 123.5000 | 118.4000 | 122.1500 |
| | SNP | 1.1818 | 1.1333 | 1.9697 | 2.3333 | 2.4000 | 2.4000 |
| | MSL | 143.2727 | 129.2333 | 136.3333 | 143.1667 | 99.8000 | 137.2000 |
| | MSD | 0.4909 | 0.4967 | 0.4955 | 0.7500 | 0.8400 | 0.4750 |
| | MSN | 13.7273 | 13.7333 | 12.9091 | 13.0000 | 11.6000 | 12.5500 |
| Ear | PL | 26.9091 | 25.0000 | 25.5758 | 31.5000 | 26.0000 | 25.5000 |
| | MEL | 21.6364 | 20.6333 | 23.8485 | 33.3333 | 20.0000 | 22.7500 |
| | MED | 49.9636 | 51.8867 | 47.6697 | 58.5167 | 25.9800 | 38.9450 |
| Yield | GWP | 49.9636 | 51.8867 | 47.6697 | 58.5167 | 25.9800 | 38.9450 |
| | TSW | 2.5455 | 2.8100 | 2.7545 | 2.5833 | 2.6200 | 2.5250 |
| | EWP | 18.5909 | 17.7100 | 25.3758 | 27.3667 | 19.7200 | 26.3850 |
| | SWP | 14.3455 | 12.9267 | 17.1697 | 17.9333 | 13.9000 | 16.9850 |

**Table 7 K-mean cluster analysis of 12 quantitative traits of 105 millet resources.**

| Traits | | Min V. | Freqency | Center V. | Freqency | Max V. | Freqency |
|---|---|---|---|---|---|---|---|
| Stem | GP | 102.21 | 0.1333 | 120.08 | 0.7143 | 127.38 | 0.1524 |
| | SNP | 1.00 | 0.3524 | 2.00 | 0.4381 | 3.00 | 0.2095 |
| | MSL | 106.42 | 0.2286 | 131.20 | 0.3810 | 152.49 | 0.3905 |
| | MSD | 0.45 | 0.7333 | 0.66 | 0.2095 | 0.97 | 0.0571 |
| | MSN | 10.63 | 0.1524 | 13.21 | 0.6952 | 15.13 | 0.1524 |
| Ear | PL | 19.63 | 0.1524 | 26.01 | 0.7429 | 34.18 | 0.1048 |
| | MEL | 19.92 | 0.4952 | 24.62 | 0.4476 | 34.33 | 0.0571 |
| | MED | 0.70 | 0.0095 | 2.44 | 0.7619 | 3.36 | 0.2286 |
| Yield | GWP | 29.71 | 0.5619 | 50.79 | 0.3238 | 82.23 | 0.2000 |
| | TSW | 2.43 | 0.5333 | 2.93 | 0.4286 | 3.63 | 0.0381 |
| | EWP | 16.80 | 0.4286 | 24.75 | 0.4000 | 31.56 | 0.1714 |
| | SWP | 11.75 | 0.4381 | 16.96 | 0.4095 | 22.43 | 0.1524 |

sheath color and bristle length that were not found in the other groups. The 2nd group displayed common characteristics for all traits. The 3rd group exhibited abundant traits characteristics, including red seed color, which was not present in the other groups. The 4th group, with its six resources, had relatively simple and concentrated trait phenotypes, but also had a rare occurrence of purple bristle color. The 5th group, consisting of only five resources, had scattered trait phenotypes, with kernel color, seed coor, and shattering habit showing unique characteristics. The 6th group showed abundant trait characteristics, with lodging resistance and ear shape displaying characteristics not found in the other groups.

Further examination of the geographical distribution of resources within each group revealed that resources from Liaoning had certain advantages, particularly in Group 4, which exclusively consisted of materials from Liaoning. Resources from Group 1 were also

concentrated in Liaoning. Group 2 resources were relatively dispersed and covered almost all geographical origins. Groups 3 and 6 mainly sourced their resources from Liaoning, Beijing, and Inner Mongolia. Group 5 resources were evenly distributed from Liaoning and Jilin.

### K-mean clustering analysis

K-means clustering analysis was conducted to analyse the 12 quantitative traits, and the theoretical distribution intervals and central values of these traits are shown in Table 7. The performance of each trait for the studied resources was found to be concentrated near the central value, with more distribution around the minimum value and less around the maximum value.

Based on these results, specific resource materials with desirable performance in terms of maturity, plant height, tiller habit, ear size, and grain weight were identified. After a comprehensive evaluation, four specific resources were selected: Dungu No. 1, an early-maturing and draft small-ear and small-seed resource from Taiyuan, Shanxi Province; Xiao-li-xiang, a small-seed and early-maturing resource from Shijiazhuang, Hebei Province; Basen Shengu, a later-maturing and taller large-ear resource from Fuxin, Liaoning Province; and Yuhuanggu No. 1, a later-maturing and low-yield large-seed resource from Chifeng, Inner Mongolia.

## DISCUSSION

This study examined a total of 31 traits, which showed rich diversity in both qualitative and quantitative aspects. Among the 19 qualitative traits, traits related to plant color, such as kernel color, seed color and leaf sheath color, exhibited lower Shannon DI values, suggesting a noticeable tendency toward pigmentation, reveal in leaf-related traits are shown in green, while ear-related traits are shown in yellow. Traits representing ear characteristics, such as ear compactness, spike density, bristle length, displayed moderate DI values, indicating that the ears of local resources are primarily sparse and loose, making them prone to grain drop. The bristle length was short, and the peduncle shape was curved, with the ear shape predominantly cylindrical and spindle shaped. Traits representing plant structure, such as tiller habit, branch habit, and blooming leaf attitude, showed higher DI values, indicating greater diversity in plant structure types.

For the 12 quantitative traits examined, the DI values ranged from 1.0566 to 2.0428, and the CVs ranged from 6.34% to 45.84%. Growing period (101–118.8 d) and 1,000-seed weight (2.0–2.68 g) exhibited small DI values, suggesting strong adaptability through long-term selection and limited potential for genetic improvement. Yield traits, such as ear weight per plant (11.5–22.5 g), seed weight per plant (9.3–15.5 g) and grass weight per plant (9.2–47.0 g), which reflect plant growth capacity, exhibited relatively high CV and DI values. This indicates that these traits are influenced by multiple quantitative genes, susceptible to external conditions, and have great potential for genetic improvement. This conclusion aligns with the research findings of *Wang et al. (2009, 2021)*.

The results of the correlation analysis indicate that ear size, stem number per plant, and growing period are closely correlated with grain yield, while no direct correlation is found between traits representing plant growth status and grain yield. These findings are consistent with those of *Jia et al. (2017)* but different from those of *Yan (2010)*, which can be attributed to variations in test sites, sampling methods, and measurement indices. Correlation analysis of qualitative traits (assigned values based on phenotype) revealed a correlation between pigmentation traits. Additionally, branch habit, tiller habit, spike density, shattering habit, seedling leaf attitude and lodging resistance exhibited correlations with yield (result supplied), consistent with the results of the diversity analysis. Since the correlation of qualitative traits is completed by assigning values on each phenotype, the influence of human factors is great, the analysis results are for reference only.

PCA revealed seven PCs. The 1st, 2nd, 3rd, 4th, and 6th PCs focus on yield and explain the associations between yield and main stem growth ability, main ear length, and growth period. The 5th and 7th PCs highlight the relationship between main stem growth ability and main ear length. In the breeding process, it is important to consider the contribution rate of each PC and the breeding goal comprehensively.

GRA based on seed weight per plant demonstrated that ear size and ear-bearing capacity are the main factors limiting yield, followed by growth period traits.

Overall, the results of correlation analysis, PCA, and GRA were consistent, suggesting that high-yield lines are expected to have more tillers and branches, larger and well-formed ears, and lodging resistance. It is worth noting that 1,000-seed weight does not directly influence yield, possibly due to millet's seed shatter characteristics and the presence of immature seeds at harvest. These findings are similar to those obtained by *Jia et al. (2021)* in their study on Tartary buckwheat, where the peduncle length was found to be proportional to the growing period, making it a potential indicator of maturity.

A systematic cluster analysis was conducted on all 31 traits, resulting in the division of 105 millet materials into six groups at a Euclidean distance of 17.09. Each group exhibited distinct phenotypic characteristics and showed certain geographical distribution tendencies. According to previous studies on crops such as adzpea (*Pu et al., 2003*) and cotton (*Xu et al., 2017*), the characteristics of these groups were found to be correlated with the natural climate conditions of their original source areas, which could be categorized using cluster analysis. The geographical distribution tendencies among groups were not prominently observed in this experiment, due to the frequent introduction of resources between regions and intermixing of bloodlines.

In addition, K-means cluster analysis was performed on 12 quantitative traits. Theoretical distribution intervals and central values were calculated to identify resource materials with specific traits (*Wang et al., 2019*). Furthermore, four specific resources were identified: one with early maturity and draft, small ear size, and small seeds; one with small seeds and early maturity; one with later maturity, taller and larger ears; and one with later maturity, lower yields, and larger seeds.

Crop phenotypes are genetically determined, and environmental factors such as temperature, humidity, soil quality, and cultivation conditions exert significant influence on them. This study solely relied on phenotype for evaluating millet resources, leading to certain limitations in the research findings. Subsequent steps will involve biotechnological research aimed at integrating genetic determinants with environmental conditions to develop more reliable evaluation methods and screening indicators. This endeavor aims to provide robust support for millet research and production.

## CONCLUSION

In conclusion, this study evaluated the phenotypic diversity of 105 millet germplasms in the Liaoning area by considering 19 qualitative traits and 12 quantitative traits. The results revealed a rich diversity of traits and complex correlations among them. Resources that possess characteristics such as upthrow seedling leaves, more tillers and branches, larger and well-formed ears, and lodging resistance prefer to higher grain yield. It was also discovered that the subear internode length could be an indicator for maturity selection. Furthermore, all resource materials were divided into six groups with different phenotypic characteristics, and the distribution interval of each quantitative character was determined. Four specific resources, namely, Dungu No. 1, Xiao-li-xiang, Basen Shengu, and Yuhuanggu No. 1, were identified for further breeding and practical applications.

### Funding

This work was supported by the General Undergraduate Colleges and Universities Scientific Research Projects of Guizhou Province (Youth Programs): Research on Inclusive Green Development Evaluation and System Coupling Coordination Mechanism of Urban Agglomeration in Central Guizhou (Grant number: QianJiaoJi[2022]165); and the Philosophy and Social Science Planning Project of Guizhou Province in 2021: Research on the Institutional Mechanism of Financial Support for High-Quality Agricultural Development in Guizhou (Project No. 21GZYB08). The funders had no role in study design, data collection and analysis, decision to publish, or preparation of the manuscript.

### Grant Disclosures

The following grant information was disclosed by the authors:
General Undergraduate Colleges and Universities Scientific Research Projects of Guizhou Province (Youth Programs): Research on Inclusive Green Development Evaluation and System Coupling Coordination Mechanism of Urban Agglomeration in Central Guizhou: QianJiaoJi[2022]165.
Philosophy and Social Science Planning Project of Guizhou Province in 2021: Research on the Institutional Mechanism of Financial Support for High-Quality Agricultural Development in Guizhou: 21GZYB08.

## Competing Interests

The authors declare that they have no competing interests.

## Author Contributions

- Xintong Li conceived and designed the experiments, performed the experiments, prepared figures and/or tables, authored or reviewed drafts of the article, and approved the final draft.
- Weifeng He performed the experiments, prepared figures and/or tables, and approved the final draft.
- Honghao Wang analyzed the data, prepared figures and/or tables, and approved the final draft.
- Min Xu conceived and designed the experiments, authored or reviewed drafts of the article, and approved the final draft.

## Data Availability

The raw data is available in the Supplemental Files.

## Supplemental Information

Supplemental information for this article can be found online at http://dx.doi.org/10.7717/peerj.17871#supplemental-information.

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
