# Peer review of "Assessment and phenotypic identification of millet germplasm (Setaria italica L.) in Liaoning, China"

_PeerJ, doi:10.7717/peerj.17871_

## Round 0.1 · original submission · Major Revisions

Three reviewers have evaluated your manuscript. While one reviewer recommended rejection due to essential modifications needed, the other two reviewers recommended revision. Based on these reviews, I suggest that the authors address all the comments provided by the reviewers to improve the manuscript. Addressing these comments will significantly improve the manuscript and give it a better chance of being accepted for publication. Ensure that all suggested modifications and improvements are incorporated into the revised manuscript. Resubmit the revised manuscript along with a detailed response to the reviewers, explaining how each comment was addressed.

We look forward to receiving your revised manuscript.

·

Basic reporting

The study aims to evaluate the phenotypic diversity of 105 millet germplasms in the
Liaoning area by considering 19 qualitative traits and 12 quantitative traits to support millet’s collection, preservation and innovation. Also, the authors employed various statistical analysis packages to assess the diversity among the millet population. The findings revealed
a considerable diversity of the studied traits and complex correlations among them.

Experimental design

Relatively well elucidated, however it needs more clarifications

Validity of the findings

Key Findings:
• The tested traits had abundant diversity and complex correlations among them with 7 principal extracted.
• Through systematic cluster, the resources were categorized into six groups at Euclidean distance of 17.09.

Reviewer 2 ·

Basic reporting

The manuscript is well written and interesting

Experimental design

The study demonstrates a good experimental design.

Validity of the findings

No comments

Additional comments

The paper titled "Assessment and phenotypic identiûcation of millet germplasm
(Setaria italica [L.]) in Liaoning, China": The study provides valuable insights into the phenotypic diversity of millet germplasm in Liaoning, China. The findings can be instrumental in breeding programs aimed at developing improved millet varieties.
-Comments and Suggestions for Authors
In title, (Setaria italica [L.]) could be written as (Setaria italica L.)
Abstract, the abstract is well written.
Introduction, the introduction could benefit from a smoother transition between the general information on millet and the specific focus on Liaoning Province. Perhaps a sentence mentioning the importance of studying germplasm resources at a regional level could bridge the gap.
In materials and methods, consider adding a table summarizing the 31 traits measured (quantitative and qualitative) could enhance readability.
Line 63: "analysed" can be changed to "analyzed" (US spelling).
In results section, line 69, Results and analysis (this title usually written as "Results")
- Line 75: "a focus on" can be replaced with "focused on" for better flow.
- Line 133: "directly represent" can be replaced with "contribute most to".
- line 167: In the analysis of 12 quantitative traits in Table 6, the qualities of each group were also compared." This sentence could be rephrased for better clarity as the following:
"Table 6 also compares the quality traits for each group identified in the cluster analysis."
- Revise the numbering of subtitles in results section
-The discussion section effectively summarizes the findings and their implications for breeding.
-Some sentences could be rephrased for better clarity. For example, sentence in lines 220- 224 can be rephrased as: "Correlation analysis of qualitative traits (assigned values based on phenotype) revealed a correlation between pigmentation traits. Additionally, black hull (BH), plant height (TH), spike density (SD), shattering habit (SH), seedling leaf attitude (SC), and lodging resistance (LR) showed correlations with yield, consistent with the results of the diversity analysis."
In conclusion, consider replacing "rich diversity" with more specific details like "high variation in tillering ability and ear size.
If possible, mention the range of observed values for key traits related to high yield.
Briefly mention potential future research directions based on the findings.

Reviewer 3 ·

Basic reporting

The paper entitled "Assessment and phenotypic identification of millet germplasm (Setaria italica [L.]) in Liaoning, China'' deal with an important issue, phenotypic identification of many millet accessions. This work was carried out over two years evaluating 105 accessions for 31 traits. I appreciate the effort deployed by the authors to realize the current study and to get huge information for different categories of traits.
Nevertheless, the paper requires a profound English editing and improvement in the paper structure and content.

Experimental design

The experiment design was clear however the method of data collection was not presented. a set of statistical analyses was used, which seems to burden the text. Also, the analyses was not explained how the data from the two cropping seasons were handled especially for quantitative traits which could significantly affected by years resulting even in significant interactions among accessions, although ANOVA analysis was not included.
With such number of accessions, working with groups (after clustering) should simplify the results, especially with multivariate analyses (PCA Bipolt).

Validity of the findings

'no comment'

Additional comments

I recommend that the paper should be re-submitted after profound editing in writing, analyses and results.
In the attached document some suggestions and corrections that might help to improve the paper.

Annotated reviews are not available for download in order to protect the identity of reviewers who chose to remain anonymous.

---

## Round 0.2 · accepted · Accept

Dear Authors,

The reviewers have noted substantial improvements in your manuscript based on their previous comments. Given these enhancements, the manuscript is now suitable for publication.

Best regards,
Elsayed Mansour


·

Basic reporting

It was my pleasure to review the article entitled: Assessment and phenotypic identification of millet germplasm (Setaria italica L.) in Liaoning, China
The authors made a substantial improvement in the manuscript and took good care of the appointed comments. I believe that the manuscript is now ready for publication in PeerJ Journal.

Experimental design

Much more improved

Validity of the findings

This research provides valuable insights into the diversity of millet in Liaoning, China using 31 traits and multiple statistical analysis packages. Further investigations are required to understand the mechanisms beyond this diversity; genetic diversity or just environmental.

Reviewer 2 ·

Basic reporting

no comment

Experimental design

no comment

Validity of the findings

no comment

Additional comments

The authors have made the changes I suggested in the last review. I recommend its publication in this journal.